# Biomechanical Behavior Characterization and Constitutive Models of Porcine Trabecular Tibiae

**DOI:** 10.3390/biology10060532

**Published:** 2021-06-15

**Authors:** Covadonga Quintana-Barcia, Cristina Rodríguez, Guillermo Álvarez, Antonio Maestro

**Affiliations:** 1SIMUMECAMAT Research Group, Universidad de Oviedo, 33203 Gijón, Spain; cristina@uniovi.es (C.R.); alvarezdguillermo@uniovi.es (G.Á.); maestroantonio@uniovi.es (A.M.); 2Hospital Begoña, c/Pablo Iglesias, 92, 33203 Gijón, Spain; 3Head of Medical Services of the Real Sporting de Gijón SAD, Av. Mareo S/N, 33214 Gijón, Spain

**Keywords:** porcine trabecular bone, uniaxial compression test, confined compression test, material behavior constitutive model, FEM

## Abstract

**Simple Summary:**

In surgery, when it comes to repairing a trauma injury, there are many variables that must be taken into account. For this reason, to study the possible effects of modifying any of the variables, it is necessary to approach the problem using numerical methods. In this work, the behavior of porcine trabecular bone, which is the most involved in this type of surgery, is experimentally analyzed to obtain constitutive models of behavior when using alternative techniques that simulate reality, such as the finite element method. Experimental compression tests were carried out, obtaining the mechanical properties of the material and the most suitable models were defined. The predictions of these models have been compared with the experimental results, thus choosing the most suitable one.

**Abstract:**

Customizing any trauma surgery requires prior planning by surgeons. Nowadays, the use of numerical tools is increasingly needed to facilitate this planning. The success of this analysis begins with the definition of all the mechanical constitutive models of the materials implied. Our target is the trabecular bone because almost all trauma surgeries are closely related to it. This work focuses on the experimental characterization of porcine trabecular tibiae and defining its best constitutive model. Therefore, different types of compression tests were performed with tibia samples. Once the potential constitutive models were defined, stress–strain state from numerical approaches were compared with the corresponding experimental results. Experimental results from uniaxial compression tests showed than trabecular bone exhibits clear anisotropy with more stiffness and strength when it is loaded in the tibia longitudinal direction. Results from confined compression tests confirmed that the plastic behavior of trabecular bone depends on the hydrostatic and deviatoric invariants, so an alternative formulation (crushable foam volumetric (CFV)) has been proposed to describe its behavior. A new method to obtain CFV characteristic parameters has been developed and validated. Predictions of the CFV model better describe trabecular bone mechanical behavior under confined conditions. In other cases, classical plasticity formulations work better.

## 1. Introduction

The current trend in traumatology is to personalize all the surgeries for each patient. That is why most trauma surgeries require prior planning in which the unique and personal strategy to be followed by surgeons with that particular patient is decided. In this planning, the elements that must be introduced in the surgery, as magnetic resonance imaging (MRI) measures, inclination angles and lengths of the grafts [1], the necessary medical instruments, the biological parts to be repaired or replaced, and the osseointegration [2] of the different elements involved in surgery are taken into account. Nowadays, it is becoming essential to work in a multidisciplinary way. That is, we have to mix this prior planning with numerical tools [3] that allow the surgeons to study the specific case of each patient and adapt the particular surgery to their physical characteristics (quality of bone, age, sex, anatomical area). Thus, each patient can be treated individually.

However, the success of any numerical approach will be based on how precise the definition of the material constitutive models is that will represent the mechanical behavior of the parts involved (biological and non-biological materials) into the numerical model [4]. For this purpose, this research needs to start with a huge experimental procedure using different test methodologies in order to describe the biomechanical behavior of the materials properly.

Particularly, this work is focused on the biomechanical analysis of a trabecular bone material model, which will be the main support of many reconstruction surgeries in traumatology (Anterior Cruciate Ligament (ACL), hip, knee…). For example, in ACL reconstruction, the trabecular bone is responsible for tendon and screw accommodation [5]. Its mechanical response is crucial in the success of the ACL reconstruction, because it is the part of the knee joint that will support the new stress state caused by the introduction of the tendon that replaces the injured ligament and the interference screw inside the tibia tunnel.

Bones are made up of cortical bone, which consists of osteons and lamellar (made by collagen fibers) and trabecular bone, which is made up of randomly oriented trabeculae. This second part of the bone is a material with tremendously complex behavior, which has been analyzed by many authors [6,7,8,9]. Furthermore, mechanical properties of this material are affected by many factors such as anatomic localization [7,10,11], the bone stress state, its mineral density [6], sex, age [9,12], and, of course, the patient’s health. Thus, the bibliography contains a huge variety of results for the mechanical parameters that characterize the bone. Hence, for an accurate analysis, correct experimental characterization of this particular bone is vital. Subsequently, these results can be used to calibrate the material in a numerical model.

There are many types of tests used to characterize this kind of bone: from macroscopic mechanical tests (tensile, compression, shear, etc.) to microscopic tests like nanoindentation [13,14,15,16]. In addition, there are different types of indirect tests such as computerized tomography (CT) [17,18], ultrasounds [19], etc. Nevertheless, comparing the results obtained using the same experimental procedure, there are many differences between authors [20,21,22]. Furthermore, most of the analyses describe only the elastic properties of the material, when it is known that the behavior of this bone is much more complex [23]. Thus, while the response of trabecular bone to tensile stress is almost linear elastic until fracture, the response to compression stress is more like the behavior of some foams or other cellular materials; once it reaches its yield strength (higher than its tensile yield strength), elastic bending of cell walls occurs. This is followed by material yield, causing the collapse of the trabeculae under constant load. This material shows an inelastic region where a stress plateau near horizontal stress represents the cell wall damage. Finally, there is a strong hardening. This behavior does not correspond to conventional models of plastic deformation, making it necessary to find the most appropriate model for its description. There are different mathematical models that have been used to characterize the behavior of the trabecular bone numerically. The models that have been considered are: isotropic linear elastic model, elastic-plastic and orthotropic model and finally, all the elastic-plastic and isotropic models in the literature (von Mises, Drucker–Prager [24], Mohr–Coulomb and crushable foam [25]). Although the anisotropic behavior of trabecular bone is known [8], an isotropic constitutive model is preferable to the orthotropic model because of the reduced computational cost. It must also be taken into account that trabecular bone is naturally confined by cortical bone. That is, its behavior is influenced not only by the response of the deviatoric invariant but also the hydrostatic invariant. crushable foam plasticity with volumetric hardening was considered as the best method to describe the behavior of tibial trabecular bone. This formulation describes the different behaviors of the material under tensile and compressive loads. Furthermore, it is known that cellular materials subjected to compressive loads suffer volumetric deformation because of the buckling of cell walls [26,27,28], thus this method seems to be the best option to represent trabecular bone behavior. One disadvantage of this constitutive model is that the determination and definition of its characteristic parameters (*K* and *K_t_*) is a very complicated task because subjecting trabecular bone samples to uniaxial tensile tests is a tremendously difficult task.

Accordingly, this paper addresses the mechanical characterization of trabecular bone from porcine tibiae. For this purpose, after a large number of experimental tests, the most appropriate constitutive model is analyzed and defined. Thus, once the constitutive model is chosen, a new method, which defines the coefficients necessary for their implementation in a finite element code, is proposed.

This mechanical characterization needs to be fed by a large number of samples. Hence, and taking into account the difficulty of obtaining sufficient human tissue for experimental characterization, we decided to work with material of porcine origin, knowing that the musculoskeletal system in this species behaves in a similar way to that of humans [5,29].

## 2. Constitutive Models of Trabecular Bone

Experimental validation of a constitutive model for trabecular bone under different loading configurations is very difficult. To date, computational models have been used to reproduce the mechanical behavior of trabecular bone, most of them under uniaxial compression loading. In addition to linear elastic material models, some non-linear elastic formulations have been used in order to reproduce the experimental behavior of this material. Moreover, verification of confined compression loading conditions are necessary as a representation of typical multiaxial physiological loading conditions of trabecular bone by cortical bone.

In the present work, Cauchy stress, σ, is the primary measure of stress and is defined as the rate between force and initial area (reference configuration). From previous studies, it is known that a continuum constitutive plasticity formulation is needed for describing trabecular bone inelastic deformation under complex loading and boundary conditions. The common assumption of elastic-plastic formulations is that the deformation can be expressed by two different components: an elastic part that is reversible and a plastic part, which is permanent. In the elastic region, the relationship between stress and strain is linear and can be represented using Hooke’s Law:(1)$$\sigma =D^{el}\cdot \varepsilon^{el}$$
where *σ* is the stress tensor, $D^{el}$ is a fourth order tensor of elastic modulus and Poisson’s ratio and $\varepsilon^{el}$ is the strain tensor of the material. The three-dimensional stress state of a material can be represented by the stress tensor, $\sigma_{ij}$:(2)$$\sigma_{ij}=\left(\begin{array}{ccc} \sigma_{11} & \sigma_{12} & \sigma_{13}\\ \sigma_{21} & \sigma_{22} & \sigma_{23}\\ \sigma_{31} & \sigma_{32} & \sigma_{33} \end{array}\right)$$

For plasticity theory, it is common to express the stress tensor in terms of the deviatoric stress tensor, *S*, and the hydrostatic stress tensor ($\sigma_{ij}^{0}$) as:(3)$$\sigma_{ij}=S_{ij}+\sigma_{ij}^{0}=S_{ij}+p\delta_{ij}$$
where the hydrostatic stress tensor, $\sigma_{ij}^{0}$, is the part of the stress which controls changes in volume and can be represented by the Kronecker tensor,$\;\delta_{ij}$, and the equivalent pressure stress (hydrostatic pressure), *p*, defined as:(4)$$p=\frac{1}{3}\sigma_{ii}=\frac{\sigma_{11}+\sigma_{22}+\sigma_{33}}{3}$$

The deviatoric stress tensor,$\;S_{ij}$, controls the distortion and can be calculated from:(5)$$S_{ij}=\sigma_{ij}-p\delta_{ij}$$

Classical plasticity theories assume that plastic deformation only depends on the components of the deviatoric stress tensor, and its second invariant (Equation (7)) is usually used to describe them.

The invariants of the deviatoric stress tensor are:(6)$$J_{1}=trace\;\left(S_{ij}\right)=S_{11}+S_{22}+S_{33}=0$$
(7)$$J_{2}=\frac{1}{2}S_{ij}S_{ij}=S_{12}^{2}+S_{13}^{2}+S_{23}^{2}-S_{11}S_{22}-S_{22}S_{33}-S_{11}S_{33}$$
(8)$$J_{3}=det\left(S_{ij}\right)$$

Thus, the $J_{2}$ invariant describes the von Mises equivalent stress, *q* (Equation (9)), which is normally used in the elastoplastic analysis of materials (i.e., von Mises and Hill) when classical theories are applied.
(9)$$q=\sqrt{\frac{1}{2}\left[{\left(\sigma_{11}-\sigma_{22}\right)}^{2}+{\left(\sigma_{22}-\sigma_{33}\right)}^{2}+{\left(\sigma_{33}-\sigma_{11}\right)}^{2}\right]}=\sqrt{\frac{3}{2}J_{2}}$$

An isotropic and symmetric yield behavior between compression and tension is defined by the von Mises plasticity formulation. For this plasticity formulation, plastic yielding is considered to be independent of the equivalent pressure stress, *p*, and yielding starts when the von Misses equivalent stress attaches the material yield stress value. Thus, the yield criterion, *F*, is defined as *F* = *q − σy* = 0. The isotropic hardening data for this model is defined from uniaxial compression tests. Furthermore, the anisotropic Hill plasticity formulation is a generalization of the isotropic von Mises plasticity formulation. Hence, yielding is again independent of the equivalent pressure stress, *p*, and it is not possible to differentiate between tension and compression behavior. The plastic anisotropic behavior is defined similarly to the von Mises model, but in this case, using the corresponding values of yield stress in the three principal directions.

For this reason, the classical plasticity theories should only be applied when plastic behavior does not depend on the volumetric material response, which seems not to be the case of the trabecular bone according to some authors [30,31]. In agreement with these researchers, trabecular bone behavior does not only depend on deviatoric stress tensor, *q*, but also on the hydrostatic one, *p*, and this particular behavior could be based on the characteristic cellular structure of this type of bone. Few possible constitutive models propose the dependency of the plastic behavior on both hydrostatic and deviatoric tensor components [24,25,31,32,33]. Among them, the best proposals to describe the behavior of trabecular bone, very similar to the exhibited by some foams, are those that have an elliptical yield surface on the meridional (*q*–*p*) stress plane. This is the case of constitutive models known as “crushable foam”, which are also included in the finite element software ABAQUS to be used in the numerical analysis of the ACL reconstruction.

In the ABAQUS library, there are two types of crushable foam plasticity formulations: isotropic hardening and volumetric hardening. The crushable foam formulation with isotropic hardening is the best option to describe materials whose isotropic yield surface is an ellipse centered on the meridional (*q*–*p*) stress plane origin. It assumes symmetrical behavior under hydrostatic tensile and hydrostatic compression, maintaining it as plasticization grows. Knowing that trabecular bone behavior under traction loads is lower than under compression, with rates of 0.6–0.9 [34] between tension and compression yield stresses, the isotropic hardening model does not seem to be the best choice, pointing to the volumetric hardening model as the more suitable option.

The crushable foam volumetric (CFV) plasticity formulation presents an isotropic yield surface with elliptical dependence of the von Mises equivalent stress on the equivalent pressure stress (Figure 1). This formulation allows differentiating between the behaviors of the material under traction loads from the one observed under compression ones. CFV plasticity formulation is motivated by the fact that cellular foams generally have different yield stress values under hydrostatic compression than under hydrostatic tension [30]. However, the great difficulty to obtain the two hydrostatic pressure condition points ($p_{t}^{0}$ and $p_{c}^{0})$ that will define the ellipse limits creates on the need to develop a new methodology in order to obtain the same ellipse with three different experimental points. In this case, confined compression tests will be very interesting in the obtaining of the representative ellipse in the *q*–*p* stress plane.

Therefore, it seems clear that we need to know which constitutive material model will be the most suitable option to represent the trabecular bone behavior, not only in terms of the capability of reproducing the experimental results but also with the aim of reducing computational cost of the finite element calculus as much as possible.

Then, the first step will be the mechanical characterization of trabecular bone. Two different loading configurations were used to define both classical plasticity formulations (von Mises and Hill) and CFV formulation: uniaxial and confined compression tests. Once the plasticity formulation parameters are obtained, an extra experimental test (roller indentation) will be done to validate the different models’ approaches.

## 3. Materials and Methods

### 3.1. Materials

Trabecular bones from porcine tibiae were used in this work. They were harvested while slaughtering the animals, in order to have a similar morphology. Knees from adult pigs (*n* = 10; estimated body weight of 85 kg) were obtained from a public slaughterhouse. The animals were slaughtered following the European Community normative for animal welfare. Stringent visual quality control was performed on each bone. Immediately after slaughter, the contiguous soft tissue was cleaned and a butchery saw was used to remove the specimens in transverse and longitudinal sections with respect to the longitudinal direction of the tibia (see Figure 2). The slices were frozen at −22 °C. Twenty-four hours before the experimental test, they were defrosted in a refrigerated chamber at 3–5 °C. The test specimens were obtained from the bone slices through shearing, using a specifically designed and manufactured die tool (Figure 3a). By stamping the bone slices with the die, (Figure 3b), cylindrical samples were obtained. The samples had a diameter of 10 mm and height equal to the thickness of the bone slices (Figure 3c). After this shearing process, the faces of the samples that were to be in contact with the compression testing plates underwent a careful polishing process until the geometry of the specimens was adequate for the compression tests. This geometry fitting process is extremely complex and delicate, in some occasions causing the inevitable breakage and loss of some trabeculae.

### 3.2. Experimental Characterization

Trabecular bone mechanical characterization was performed using three different tests: uniaxial compression tests, confined compression tests and roller indentation tests. All the mechanical tests were done at room temperature in an MTS® static test machine with 5 kN of load capacity, an accuracy of ±0.1 N and using a load point displacement rate of 2.5 mm/min.

Uniaxial compression tests (Figure 4a), were performed using the typical equipment for this kind of test for over 40 samples (20 in longitudinal loading and 20 in transverse loading). It consists of two rigid plates that compress the sample in one direction, allowing it to be deformed in all directions. Thus, the samples denominated “longitudinal” were subjected under compression loads in the homonymous direction of the tibia, as well as samples under “transverse” denomination. In this sense, the transverse samples come from bone slices extracted from cuts parallel to the longitudinal axis of the tibia, while the longitudinal samples come from perpendicular cuts to said axis. Displacement was measured by an MTS® crack opening displacement (COD) extensometer (accuracy of ±0.24 µm) placed between the two compression plates (Figure 4).

In order to measure trabecular bone deformation with more precision, a digital image correlation device (DIC) was used in 5 of the tests. The DIC device used was a GOM® ARAMIS 5M with 50 mm focal distance lenses and a calibration panel CQCCP20 30 × 24. This permits a measurement zone of 35 × 29 mm^2^. Aramis is a non-contact optical 3D deformation measuring system that analyzes, calculates and documents material deformations. It is particularly suitable for three-dimensional deformation measurements under static and dynamic load in order to analyze deformations and strain of real components. This equipment requires careful preparation of the samples; a black mottling on a white background was applied to the surface. These references were used by the equipment to identify the modifications in their positions in the specimen, which are directly related to the deformation suffered by the sample during the test. For a 3D measurement setup, two cameras are used and calibrated prior to measuring. Figure 5 shows the uniaxial compression test results recorded by means of DIC equipment. The use of these images makes it possible to determine the Poisson’s ratios as follows: the DIC software allows obtaining a 3D surface. A cylinder shape (in this case, only the center portion has been used due to its greater strain) can fit this surface. The relationship between the diameter differences along the test and the initial one, gives the final transverse strain. The correlation of this parameter with the longitudinal strain gives the final Poisson coefficient result. This procedure was applied to both the transverse and longitudinal bone samples.

Confined compression tests were performed using the same methodology as in the uniaxial ones but maintaining the samples in confined conditions inside the shearing device used to extract them from the bone slices. Trabecular bone, laterally confined inside the die, was axially compressed by a cylindrical punch, with a diameter identical to the internal diameter of the die (Figure 4b). In this case, only the COD displacement device could be used to obtain the strain measures and only “transverse” samples were used. This type of test was carried out using 10 specimens whose load axis coincided with the tibia transverse direction. The lack of biological material prompted the decision of using only transverse for loading, since this direction is likely to show lower values of stiffness and strength [35,36].

At this point, it is necessary to clarify that uniaxial compression tests are those normally used in the trabecular bone characterization when it is assumed that their behavior can be described by means of classical plastic theories. However, the execution of confined compression tests was focused on analyzing whether the mechanical behavior of the trabecular bone could be adjusted to a more complex material model. That is the “crushable foam”, which will be described later. The bibliographic references [30,31] pointed it out as one of the most appropriate to describe the behavior of this type of bone. In this way, once the confined compression tests have been carried out, and the parameters that describe the material model have been obtained, it is necessary to execute a third mechanical test that will allow validation of the material model with numerical tools. That is, comparing the experimental curves of the test and those obtained through the numerical model of the same experimental procedure. The test selected was roller indentation.

In roller indentation tests, a prismatic sample of trabecular bone was subjected to the action of a steel cylinder that compressed the central part of one of the faces, finding the opposite face completely supported on a compression plate (Figure 6).

These tests were performed over six trabecular bone specimens of similar measurements (approximately 35 × 15 × 10 mm^3^) that were extracted from longitudinal slabs of the tibia, so that their longitudinal direction coincided with that of the bone, thus ensuring that the direction of the load was always transverse.

This test was carried out with the help of DIC techniques (Aramis), such as uniaxial compression tests, which will allow verifying how the specimen deforms as the load increases in a much more precise way (Figure 6b).

In all cases, the value of the load versus displacement was collected using the specific test software of the equipment.

### 3.3. Finite Element Modelling

Both types of compression tests (uniaxial and confined) have been modelled using a 2D axisymmetric model. Cylindrical samples with rectangular sections of 10 mm in height and a radius of 5 mm were used to simulate the two types of tests, using the necessary boundary conditions for each case. Two hundred quadratic quadrilateral elements with full integration (CAX8) were used to mesh the samples. The mechanical properties used in the FE model were obtained experimentally from the uniaxial compression test of the tibiae transverse samples.

The roller indentation test was also modelled using a 2D axisymmetric model. Prismatic samples with rectangular sections of 12 × 19 mm^2^ were subjected to the action of a roller punch of diameter 12 mm (modelled as a rigid body). Samples were defined with the parameters that best described the material model in each case. The mesh consisted of 1650 type CPE4 (bilinear 4-node plane strain elements). Boundary conditions are those corresponding to the experimental procedure, preventing axial displacement and rotation of the specimen, as well as radial displacement of one of the base points. The movement of the roller punch will be restricted in such a way that only its axial movement is allowed, adding interaction properties for the contact between the roller and the specimen with a friction coefficient of 0.2. The simulation consists of applying a certain axial distance to the roller and obtaining the load-displacement curves of the reference point that represents it.

## 4. Results

### 4.1. Uniaxial Compression Tests

Figure 7a shows mean and standard deviation stress–strain curves (*σ*–*ε*) from the experimental results obtained after uniaxial compression tests of trabecular bone specimens both in the longitudinal (11 samples) and transverse direction (16 samples). The represented values correspond to the apparent stress, *σ* (using the initial cross-sectional area, *S*_0_, to calculate the stress), and the apparent strain (using the initial length of the specimen, *L*_0_, and the COD extensometer displacement data to calculate it).

Regardless of the high anisotropy of this type of bone, which exhibits higher strength values in longitudinal direction than in transverse direction samples, we can observe that in all cases, the mechanical behavior is similar to that described in Figure 7b. After an initial linear zone whose slope is the elastic modulus, *E*, a yield point is reached and, then, a plastic plateau appears, where the stress remains practically constant until the strain reaches a considerable value. At that moment, a hardening process due to bone densification takes place. Although the hardening process in the figure is not equally visible in both loading directions, the behavior is the same, with the only difference that in transverse loading direction it takes place at higher strain levels. This behavior, already observed by other researchers [30,34,35,37,38], is based on the trabecular structure of this type of bone, which makes it behave like a foam [38,39].

Based on the curve obtained, it seems evident that the mechanical behavior of the analyzed trabecular bone could be modeled as an elastic-perfectly plastic material with transverse isotropy (different behavior in the longitudinal direction but the same behavior in any direction that is contained in the transverse plane). To define this type of model, it is necessary to calculate both elastic constants (Young’s modulus, *E*, and Poisson’s ratio, ν) and the yield stress, *σ*_y_, in both principal directions.

Table 1 reflects the average values and standard deviation of these parameters obtained in the two directions analyzed. As can be seen in Figure 7b, the yield stress values were defined as the stress value on the first maximum in the *σ*–*ε* curve. The Poisson’s ratio was obtained as the relation between radial and axial strains of the sample using DIC techniques for displacement measurements described above.

As other researchers have found [40,41], porcine trabecular bone shows clear anisotropy. The longitudinal stiffness is almost three times higher than transverse stiffness, and the longitudinal compression strength is double the transverse strength.

Furthermore, the high dispersion of the results from the same type of bone (tibiae) and from individuals of similar age and build is noteworthy. This dispersion is typical of this kind of bone [6,11,20,21,22], and is due to many reasons: different bone density between individuals, different position over the bone, inaccuracies in samples preparation, etc.

### 4.2. Confined Compression Tests

The apparent stress–strain curves obtained from confined compression tests are shown in Figure 8 in terms of mean and deviation values. Figure 8 also shows, as a comparison, the mean stress–strain curve representative of the uniaxial compression tests in the transverse loading direction. As in previous tests, the results obtained from the confined compression tests also show a high dispersion, its stiffness and strength are clearly higher than that obtained in uniaxial compression tests.

The average value of the initial slope in confined compression tests is 106 ± 38 MPa, while the average value of the yield stress is 5.1 ± 1.5 MPa (almost twice the value obtained in uniaxial compression tests).

### 4.3. Roller Indentation Tests

Figure 9 shows the load-displacement curves of the loading cylinder, obtained after roller indentation tests of porcine trabecular bone specimens. Now, the differences between the experimental curves could be due to small variations in the dimensions of the specimens (note that the graph is load–displacement) instead of the characteristics of the bone itself.

## 5. Definition of Trabecular Bone Constitutive Models

As mentioned before, stress–strain curves obtained from experimental tests seem to indicate a clear pattern of behavior. That is very similar to that exhibited by other porous materials like foams. In this sense, it is probable that the most appropriate constitutive model is the CFV, but the use of this model in a numerical evaluation can manifest two problems: the difficulty of obtaining its characteristic parameters and the large calculus time (computational cost) that is needed to obtain a numerical solution. This problem does not appear using classical plasticity formulae such as von Mises or Hill, whose parameters can easily be obtained and the calculus time is shorter (particularly the von Mises formula). Thus, starting from the experimental results from compression tests, we will define three material constitutive models: von Mises, Hill and CFV. Once the material constitutive model is defined and implemented in a finite element code, we will compare the predictions of each model when they are applied to describe the behavior of the trabecular bone subjected to a different mechanical test with different loading requests the roller indentation test produces.

The transverse isotropy showed by the trabecular bone under uniaxial compression loads, pointed to the Hill model as the most appropriate when classical theories of plasticity are chosen. To define this model, the characteristic elastic-perfectly plastic parameters in the longitudinal (*E*_L_, ν_LT_ and *σ*_yL_) and transverse direction (E_T_, ν_TL_ and *σ*_yT_) must be defined. These parameters are taken from the results of uniaxial compression tests (see Table 1). At this point, it is necessary to clarify that when Hill’s formula is applied, the material orientation with respect to the loading axis must be defined, and this fact could be a problem when the orientation of the trabeculae is not clear.

In order to confront this problem, a good solution could be assuming the behavior of the material is isotropic, using the properties of the weakest direction. Based on this, the von Mises model using the transverse elastic-perfectly plastic parameters (*E*_T_, ν_TL_ and *σ*_yT_) showed in Table 1, was defined.

Regarding the CFV model, which assumes the behavior of the material is isotropic, as can be seen in Figure 1, the proper definition of its elliptical plasticity surface will need at least three different points. Two of these optimal points are the values of yield stress in hydrostatic traction ($-p_{t}^{0}$) and compression ($p_{c}^{0}$) conditions. These two parameters define the ellipse (Figure 1) ends over the hydrostatic p axis. The third point necessary to define the ellipse is usually the uniaxial compression yield stress ($\sigma_{c}^{0}$). Although it depends on the finite element code used, with these parameters, the CFV model can be completely defined. In the case of using ABAQUS code, the coefficients K and $K_{t}$, will be defined as:(10)$$K=\frac{\sigma_{c}^{0}}{p_{c}^{0}}$$
(11)$$K_{t}=\frac{p_{t}^{0}}{p_{c}^{0}}$$

Nonetheless, the great difficulty of obtaining tensile and compressive yield stresses in hydrostatic conditions has caused many authors to use alternative methods to define the shape of the yield ellipse [30,42]. Kelly [30] assumes $K_{t}=1$ and with the help of a finite element model and an inverse method, this author chooses the value of *K* that best fits the numerical results with the experimental ones. Moreover, [42] define the model using alternative points obtained experimentally from different loading conditions (a large amount of synthetic foam material was needed).

In this work, an intermediate solution is chosen. On one hand, in order to define the CFV yield ellipse, two points were obtained experimentally (see Figure 10): the yield stress in uniaxial compression conditions ($\sigma_{c}^{0}$/3, $\sigma_{c}^{0}$) and the yield stress in confined compression conditions (point *q_conf_*). The position of these points in the yield ellipse can easily be obtained by applying the corresponding contour conditions. When uniaxial compression is applied, *σ*_1_ = $\sigma_{c}^{0}$, and *σ*_2_ = *σ*_3_ = 0, so *p* =$\;\sigma_{c}^{0}$/3 (Equation (4)) and *q* =$\;\sigma_{c}^{0}$ (Equation (9)). In the case of confined compression *σ*_1_ = $\sigma_{conf}^{0}$ and transverse strain (*ε*_2_ = *ε*_3_) is zero, so applying Hooke’s law (Equation (1)): *σ*_2_ = *σ*_3_ =$\;\frac{\nu }{1-\nu }\sigma_{1}$, and therefore, *q_conf_* = 1.08 *p_conf_*.

The representative point of the uniaxial compression yielding was obtained using the elastic-plastic parameters in transverse direction (see Table 1). In the case of confined compression tests, due to the large variability obtained experimentally, we decided to work with two different values of yield stress under confined conditions: $\sigma_{conf}^{0}$ = 3 MPa (Figure 10a) and $\sigma_{conf}^{0}$ = 5 MPa (Figure 10b).

The third point necessary to define the yield ellipse was defined, in this case, using the value of the yield stress in tensile conditions. This value was obtained from experimental results supported by other authors [18,30,34,43], which defined it as 60–90% of the yield stress in compression conditions. Furthermore, this point must be located at the ellipse point intersecting the straight *q* = −3*p*, i.e., the point ($-\frac{\sigma_{t}^{0}}{3},\;\sigma_{t}^{0}$) in Figure 10.

With these points, the yield ellipse is drawn (Figure 10) and it is possible to obtain both coefficients, *K* and *K_t_,* that will define the CFV material model in ABAQUS. The values of these parameters using the different combinations of possible yield stress under confined and uniaxial tension conditions are showed in Table 2. As can be seen, the value of *K* will depend fundamentally on yield stress values in confined compression conditions. *K* values are obtained below 0.4 when $\sigma_{conf}^{0}\;$= 5 MPa and above 1 (*K* = 1.2) when $\sigma_{conf}^{0}\;$= 3 MPa. *K_t_* is more dependent on the tensile yield stress value considered.

In order to choose the pair of coefficients that can better describe the CFV constitutive model, the confined compression test was simulated using the FE model previously described in Section 3.3. The numerical results obtained using each pair of *K* parameters were compared with the experimental ones. Figure 11 and Figure 12 show the numerical curves (continuous lines) obtained using the different *K* values corresponding to $\sigma_{conf}^{0}\;$= 3 MPa and $\sigma_{conf}^{0}$ = 5 MPa, respectively. These figures also show some of the curves obtained experimentally from confined compression test (dashed lines). Likewise, and in order to compare the uniaxial and confined behaviors, a representative curve of the uniaxial compression test is also drawn (thick dotted line).

In the case of using parameter values corresponding to $\sigma_{conf}^{0}$ = 3 MPa (Figure 11), it is observed that the only curve that can estimate the behavior of the confined compression experimental curves in the yield zone is the one that considers a tensile yield stress equal to 90% of compression yield stress. This curve has the same stiffness as the other numerical curves, but it estimates a more precise yielding behavior compared to that recorded in the confined compression tests. However, after reaching this stress peak, the numerical curve stops following the experimental behavior, where the material exhibits exponential hardening, and it starts following the behavior described by the uniaxial compression test. The other numerical curves barely perceive the increase in yield stress in confinement conditions, and practically exhibit uniaxial compression behavior.

In the case of using parameter values corresponding to $\sigma_{conf}^{0}\;$= 5 MPa (Figure 12), all the numerical curves are practically identical whatever the relationship between tensile and compression yield stresses used is. All the curves are now able to predict an increase in the confined yield stress. As in the previous case, beyond that point, not all the numerical cases are capable of reproducing the material behavior under confinement conditions. Instead, they return to the plastic behavior of the material under uniaxial compression loads.

Based on these results, it is intuited that this model is also not capable of faithfully reproducing the mechanical behavior of this type of bone.

Once the three constitutive models that can describe the mechanical behavior of porcine trabecular bone have been defined, their fitness has been analyzed by applying them to the numerical simulation of the roller indentation test described above.

Figure 13 shows the load-displacement curves of the roller indentation test numerically obtained by means of the three analyzed constitutive models (continuous lines). This figure also shows, representatively, one of the experimentally obtained curves (dashed line “Experimental T2”). As can be seen, the numerical curves obtained with the CFV model, both with the most appropriate parameters for $\sigma_{conf}^{0}$ = 3 MPa (*K* = 1.28, *K_t_* = 0.76) and for $\sigma_{conf}^{0}\;$= 5 MPa (*K* = 0.309, *K_t_* = 0.371), are moved away from the experimental ones. On the contrary, the results obtained when the classical constitutive models are used, and particularly with the Hill’s model, are closer to the experimental results of the roller indentation test than using the CFV model, whose predictions are very far from reproducing such behavior.

The fact that the CFV model does not predict the roller indentation test results as well as the other two constitutive models could be because the validity of the CFV model is restricted to simulate cases in which the material is working under confined conditions, as in the case of the trabecular bone in its natural conditions (inside the cortical bone). For this reason, it is advisable not to reject the possibility of using the CFV model when the case studied is under these specific conditions, such as the analysis of reconstruction of the anterior cruciate ligament.

## 6. Conclusions

Uniaxial compression tests over longitudinal and transverse porcine trabecular bone samples have clear anisotropic mechanical behavior, with higher stiffness and strength of the bone when loads are applied in the direction of the longitudinal axis of the tibia. The elastic modulus in longitudinal direction (218 MPa) is three times higher than that obtained transversally while the yield stress value (5.3 MPa) is practically double.

DIC techniques have made obtaining the Poisson’s ratio of trabecular porcine bone possible. It is 0.25 in the longitudinal direction and 0.27 in the transverse one.

The design and execution of confined compression tests has made it possible to support crushable foam with volumetric hardening as one of the constitutive models that could best describe the post-yielding behavior of porcine trabecular bone in confined conditions. Classical plasticity models, i.e., von Mises and Hill, have also been used to describe mechanical behavior of porcine trabecular tibiae.

Roller indentation tests were used in order to validate the three proposed material behavior models. The CFV model does not correctly follow the trabecular bone experimental behavior at least when subjected to these loading conditions. However, classical plasticity formulae, and particularly the Hill’s theory, are capable of predicting the behavior exhibited for the trabecular bone under the roller indentation solicitations with great precision.

## Figures and Tables

**Figure 1 biology-10-00532-f001:**
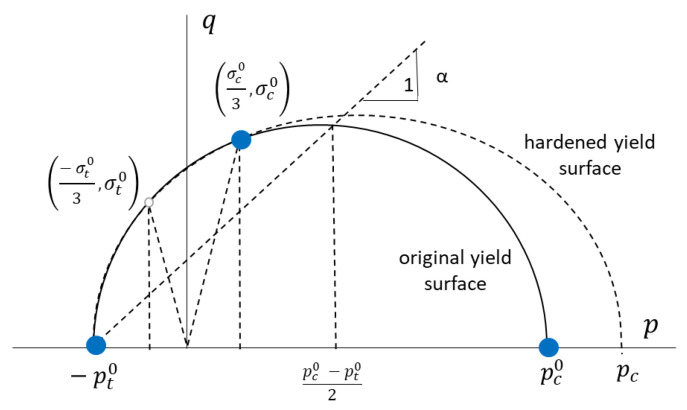
Ellipse in the meridional (*q*–*p*) stress plane of the crushable foam volumetric (CFV) model.

**Figure 2 biology-10-00532-f002:**
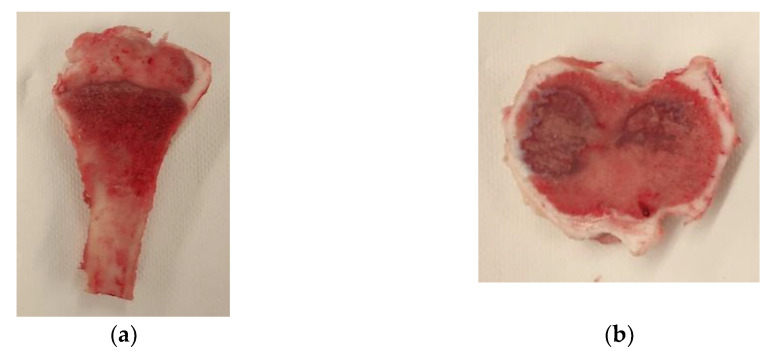
Trabecular bone slices: (**a**) for the transverse load direction; (**b**) for the longitudinal load direction.

**Figure 3 biology-10-00532-f003:**
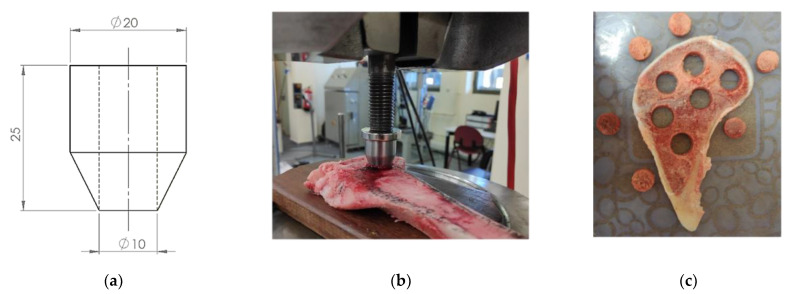
Extraction of the samples from the slices: (**a**) extraction tool (measurements in “mm”); (**b**) specimen extraction process; (**c**) extracted specimens from one slice.

**Figure 4 biology-10-00532-f004:**
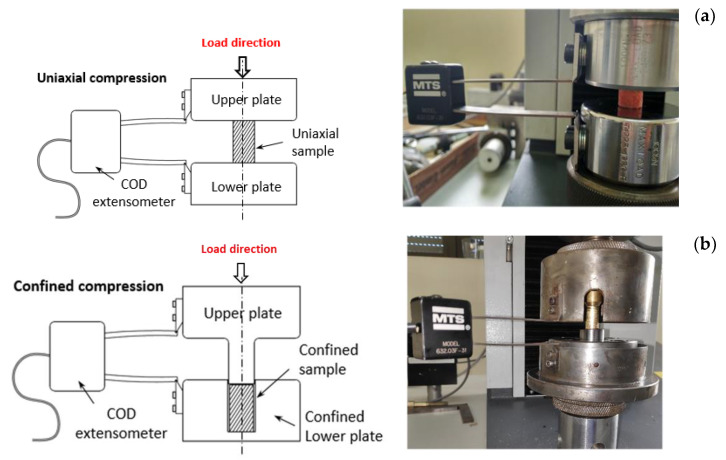
Uniaxial (**a**) and confined (**b**) compression tests configuration.

**Figure 5 biology-10-00532-f005:**
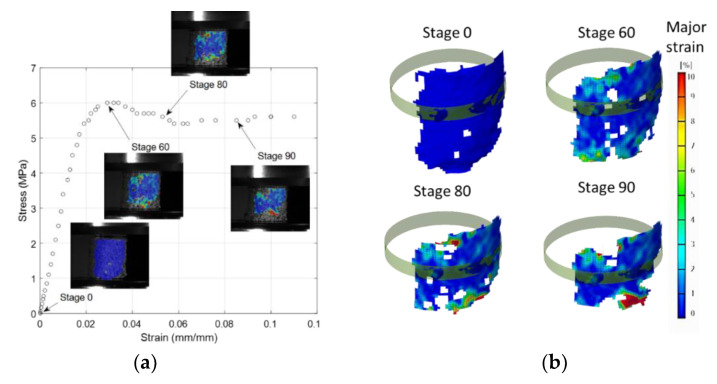
Images from a uniaxial compression test using digital image correlation (DIC) techniques: (**a**) different stages; (**b**) cylinder shape approximation.

**Figure 6 biology-10-00532-f006:**
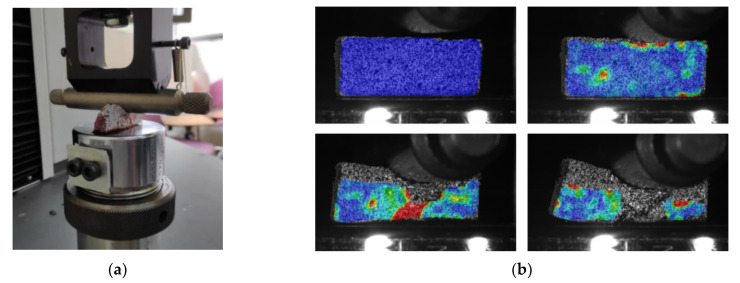
Roller indentation test: (**a**) prismatic trabecular bone sample during test; (**b**) strain measures by DIC techniques.

**Figure 7 biology-10-00532-f007:**
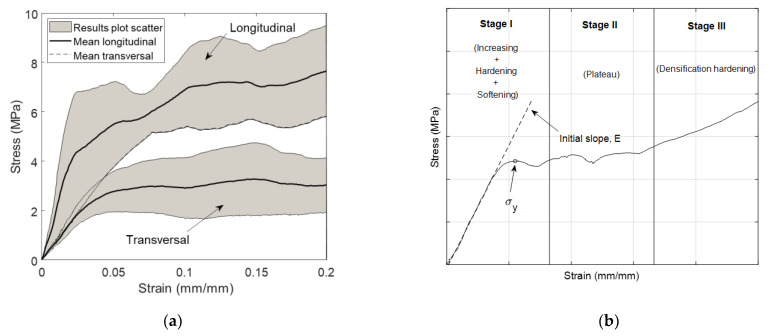
Uniaxial compression tests curve in both loading directions (**a**); example curve of the behavior of trabecular bone under uniaxial compression conditions (**b**).

**Figure 8 biology-10-00532-f008:**
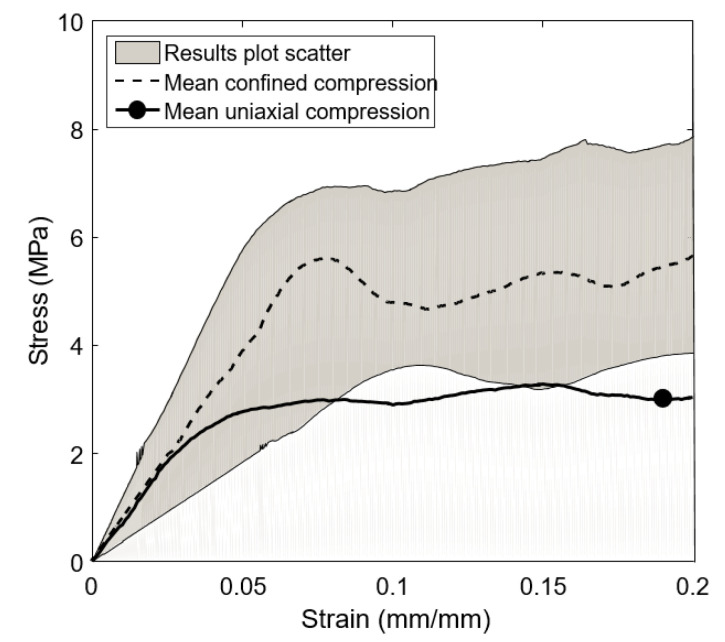
Stress–strain curves of confined compression tests vs. uniaxial compression tests.

**Figure 9 biology-10-00532-f009:**
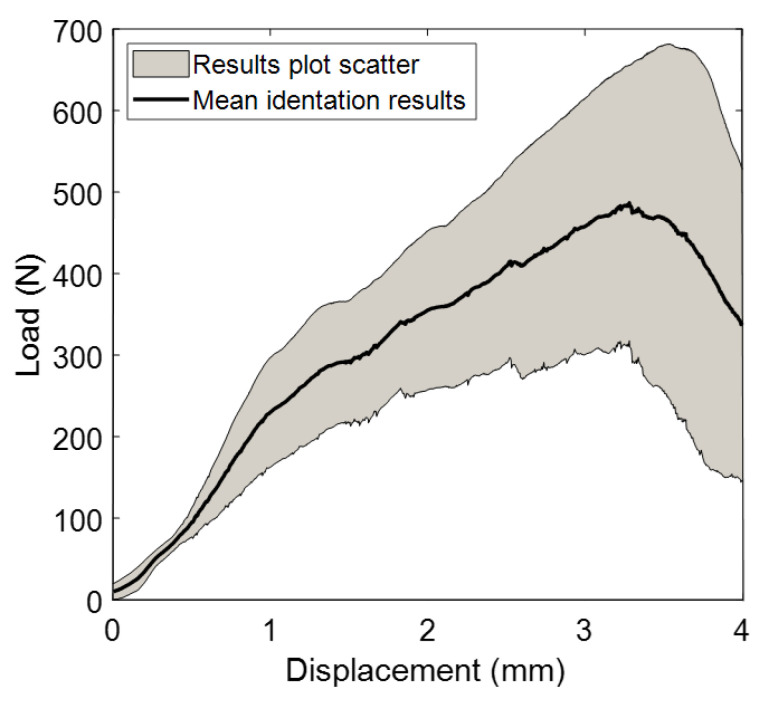
Load–displacement curves from roller indentation tests.

**Figure 10 biology-10-00532-f010:**
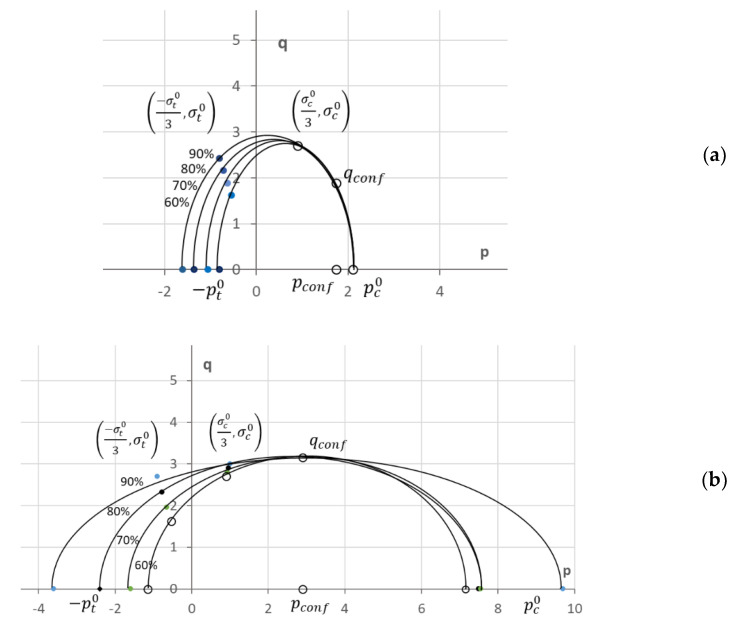
Possible ellipses of CFV constitutive model for trabecular bone with (**a**) 3 MPa and (**b**) 5 MPa.

**Figure 11 biology-10-00532-f011:**
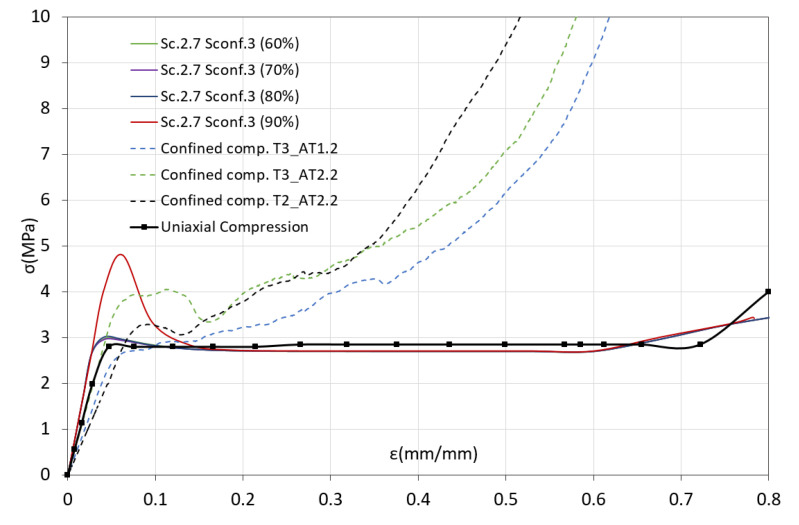
Experimental stress–strain curves (dashed line) vs. numerical (continuous line). Effect of CFV constitutive model parameters in the case of $\sigma_{conf}^{0}=3\;\mathrm{MPa}$.

**Figure 12 biology-10-00532-f012:**
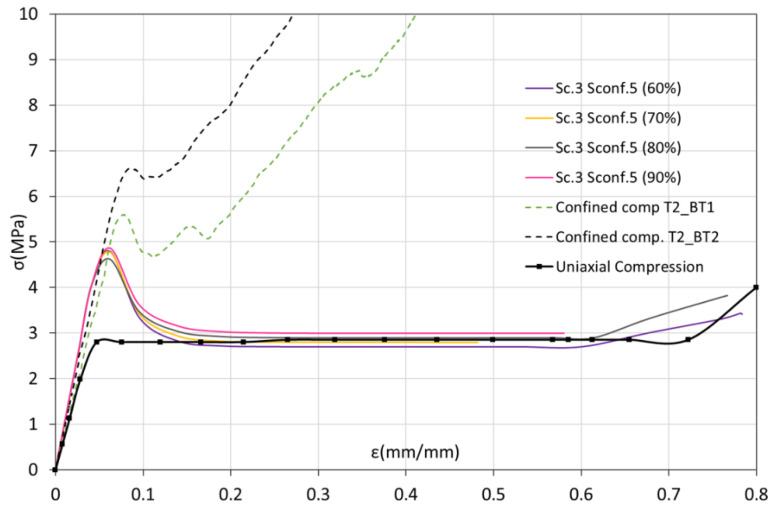
Experimental stress–strain curves (dashed line) vs. numerical (continuous line). Effect of CFV constitutive model parameters in the case of $\sigma_{conf}^{0}=5\;\mathrm{MPa}$.

**Figure 13 biology-10-00532-f013:**
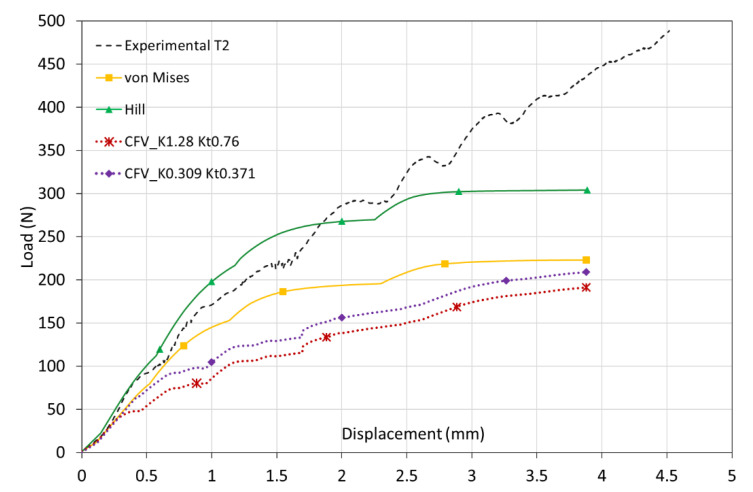
Comparison of experimental load–displacement curve with numerical of roller indentation test using CFV, von Mises and Hill’s constitutive models to represent trabecular bone behavior.

**Table 1 biology-10-00532-t001:** Characteristic mechanical values of porcine trabecular bone in both orthogonal directions.

Test	Direction	*E* (MPa)	*σ*_y_ (MPa)	ν
Uniaxial compression	Longitudinal	218 ± 134	5.3 ± 1.1	0.25 ± 0.02
	Transverse	73 ± 15	2.8 ± 0.7	0.27 ± 0.02

**Table 2 biology-10-00532-t002:** Parameters of the CFV constitutive model obtained from numerical characterization of confined compression tests.

$\sigma_{c}^{0}\;(\mathrm{MPa})$	$\sigma_{conf}^{0}\;(\mathrm{MPa})$	$\sigma_{t}^{0}\;(\mathrm{MPa})$	*K*	*K_t_*	Numerical Designation
2.7	3	0.6 $\sigma_{c}^{0}$	1.285	0.380	Sc. 2.7 Sconf.3 (60%)
		0.7 $\sigma_{c}^{0}$	1.280	0.500	Sc. 2.7 Sconf.3 (70%)
		0.8 $\sigma_{c}^{0}$	1.220	0.610	Sc. 2.7 Sconf.3 (80%)
		0.9 $\sigma_{c}^{0}$	1.280	0.760	Sc. 2.7 Sconf.3 (90%)
3	5	0.6 $\sigma_{c}^{0}$	0.377	0.160	Sc. 3 Sconf.5 (60%)
		0.7 $\sigma_{c}^{0}$	0.373	0.213	Sc. 3 Sconf.5 (70%)
		0.8 $\sigma_{c}^{0}$	0.384	0.317	Sc. 3 Sconf.5 (80%)
		0.9 $\sigma_{c}^{0}$	0.309	0.371	Sc. 3 Sconf.5 (90%)

## Data Availability

We choose to exclude this statement; data of the main values of results are included in the manuscript. At this time, the data also forms part of an ongoing study.

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
