# Peer review of "Biomechanical Behavior Characterization and Constitutive Models of Porcine Trabecular Tibiae"

_biology, 2021, doi:10.3390/biology10060532_

Round 1

Reviewer 1 Report

In this work the author approached the modeling of porcine trabecular bone, by carrying out a series of experimental tests and by using different mathematical models. The manuscript is more likely an essay on mathematical modeling of bone behavior, and the novelty of this research is not clear to the reader. Indeed, a huge of work is carried out to conclude that the best model that describes trabecular bone behavior is the standard one, and that the CFV model is not able to predict the roller indentation test results since this dos does not work under confined conditions, which could be expected.

I have other major and minor concerns:

  • The relationship between ACL reconstruction and porcine trabecular bone behavior is completely not clear. This seems a forcing to the reader.
  • Some of the results reported a SD higher that 50% of the mean value. I agree that biological samples have high variability, but in my opinion to obtain a better modeling of tissue behavior it is necessary to reduce this variability. Did the authors verify that all the samples contained only trabecular bone? Did they verify the orientation (as stated)? They could probably refer to smaller samples to be more confident on their high reproducibility. An increase in the quality of the measurements could lead to an increase in the quality of the modeling.
  • The authors stated that during tests for DIC analysis they did not hydrate the samples. Did they hydrate the samples during the other tests? Did they observed any difference in the mechanical response of the samples if hydrated or not? Please quantify this error (even if negligible).
  • The authors must improve the description of the experimental system: is the DIC system composed by two cameras? Please describe
  • The authors must improve the description of the experimental system: please provide the metrological properties of the instruments, i.e., measurement sensitivity, accuracy… for the two variables of interest (force and length)
  • Is Figure 7b from previously published work? is it only a drawing? Please clarify and properly refer
  • The behavior of the bone to transverse loading does not show the hardening process due to bone densification. Please better clarify and avoid inferences. Again, better measurements would probably help modeling.
  • Please clarify the number of samples tested from transversal and longitudinal slices for each test. This is clearly stated only for the confined test

Author Response

The authors would like to thank the reviewer for your valuable comments and suggestions. We can assure you they have been taken into account, as you will be able to see in the next version of the manuscript.

Please, see the attachment document with all the answers to your suggestions and comments. 

Reviewer 2 Report

Interesting work, needs some English editing. Pls see the attached manuscript with my comments.

Author Response

The authors would like to thank the reviewer for your valuable comments and suggestions. We can assure you they have been taken into account, as you will be able to see in the next version of the manuscript.

We upload the pdf version with your comments answered in the same pdf. 

Round 2

Reviewer 1 Report

The revised version of the manuscript has been improved accordingly to most of my suggestion.

However, I still believe that the explicit link with ACL reconstruction is forced and out of the aim of this manuscript. All the manuscripts are part of bigger research projects.

Author Response

Dear reviewer, 

We would like to thank you for your comments. We believe that your suggestion related to the link of the paper with ACL reconstruction is corrected in this new version, and we also hope it is better explained in section 1 and abstract.  
We attached the new version of the paper with "track changes".

Thank you. 

Best regards